# Endoglin and Other Angiogenesis Markers in Recurrent Varicose Veins

**DOI:** 10.3390/jpm12040528

**Published:** 2022-03-25

**Authors:** Francisco S. Lozano Sánchez, José A. Carnicero Martínez, Lucía Méndez-García, M. Begoña García-Cenador, Miguel Pericacho

**Affiliations:** 1Service of Angiology, Vascular and Endovascular Surgery, University Hospital of Salamanca (HUS), 37007 Salamanca, Spain; lozano@usal.es (F.S.L.S.); josecarnicero@hotmail.com (J.A.C.M.); 2Department of Surgery, University of Salamanca (USAL), 37008 Salamanca, Spain; 3Biomedical Research Institute (IBSAL), 37007 Salamanca, Spain; luciamengar@usal.es (L.M.-G.); pericacho@usal.es (M.P.); 4Department of Physiology and Pharmacology, University of Salamanca (USAL), 37007 Salamanca, Spain

**Keywords:** varicose veins, recurrent varicose veins, angiogenesis, endoglin

## Abstract

Background: Surgery on varicose veins (crossectomy and stripping) may lead to recurrence, with clinical and socioeconomic repercussions. The etiopathogenesis of varicose veins has yet to be fully understood. Objective: Study the expression of endoglin and other molecules involved in the neovascularisation process in patients suffering from this disease. Methods: Total of 43 patients that have undergone surgery for varicose veins (24 primary and 19 recurrent). Endoglin and other molecules were identified on the venous wall (proximal -saphenofemoral junction- and distal), via real-time RT-PCR, and in serum, via ELISA: endoglin (Eng), vascular endothelial growth factor (VEGF-A), its receptors 1 and 2 (VEGFR1 or FLT1), (VEGFR2 or FLK), and the hypoxia-inducible factor (HIF-1A). All the patients signed a consent form. Results: The recurrent group recorded a higher expression of Eng, VEGF-A, VEGFR1, and VEGFR2 at the level of proximal venous wall compared to the primary group. HIF-1A did not record any differences. As regards the determination of the distal venous wall, no markers recorded differences between the groups. Among the serum determinations, only sFLT1 recorded a significant drop among the patients with recurrent varicose veins. Conclusions: Patients with recurrent varicose veins record a higher expression of endoglin and other markers of angiogenesis in proximal veins. Endoglin in the blood (sEng) serves no apparent purpose in recurrent varicose veins.

## 1. Introduction

Varicose veins (VV) are highly prevalent in western societies [1], featuring among the top ten most common surgical procedures and involving the longest waiting lists for operations in public health systems [2,3]. All-in-all, VV constitute the perfect storm of clinical, social and economic problems; some examples are their associated complications (e.g., ulcers), a decrease in wellbeing, and the medical-legal issues they entail [4,5,6].

The scope of the problem is even greater because one out of every four patients receiving treatment for VV is suffering from the recurrent form (RVV) [7], and up to 20% of the operations on VV involve recurrence [8]. The rate of RVV fluctuates between 13% and 65%, varying according to the technique used [9], whereby following a saphenofemoral ligation it may reach 60% [10]. It is well known accordingly that repeat instances of VV surgery are technically more complex, take longer, are less successful, reduce patient satisfaction, and incur higher costs.

There are three types of RVV: (1) residual or varicose ones that were not treated in the initial intervention and which have been detected in an early check-up; they are due to tactical/technical errors; (2) true recurrences, which appear during treatment on the affected area; they may be due to tactical/technical errors or neovascularisation; and (3) new VV, which appear in untreated areas and are caused by the advance of the disease [7,9].

A sound diagnosis and proper surgical procedures can only prevent tactical/technical errors. The other causes account for 50% of RVV. The cause of the recurrence has not been identified in 10–35% of the cases; in turn, 50–70% of the recurrences are located in the saphenofemoral junction, where neovascularisation is as common as a tactical/technical error [9,11].

While neovascularisation is a frequent factor of RVV, there should be/persist a process of angiogenesis, which would mean there has been angiogenic neovascularisation and that this process has been maintained. This study, therefore, seeks to investigate the behaviour of sundry molecules (markers of angiogenesis, inflammation, and hypoxia) involved in these processes; we are focusing accordingly on endoglin, a molecule whose behaviour has not been explored in RVV.

Endoglin is a membrane glycoprotein that is expressed especially in endothelial cells. As a pro-angiogenic molecule, endoglin plays a key role in neo-angiogenesis regulation [12].

The aim is to study the possible role of the neovascularisation process as a cause of RVV by comparing endoglin and other molecule levels in a group of patients undergoing surgery for the first time with another group of patients with RVV.

## 2. Materials and Methods

A clinical study of a prospective, observational, open, controlled, and non-randomised nature has been conducted at the Angiology and Vascular Surgery service at the University Hospital in Salamanca (Spain). Figure 1 presents the research scheme.

### 2.1. Patients

Inclusion criteria: men and women aged 18–70, classification ASA I–II, with VV, CEAP classification 2–6, diagnosed by eco-Doppler with insufficiency of the saphenofemoral junction or neo-junction (according to groups), and subject to a crossectomy (saphenofemoral ligation/section) and stripping of the great saphenous vein (GSV), and who give their written consent to take part in the study. All patients with RVV; the GSV must have previously been removed by crossectomy + stripping.

Exclusion criteria: pregnant women or those that have given birth over the past 12 months, obesity (body mass index-BMI > 30 kg/m^2^), traumatism or surgery over the past six months, serious or chronic inflammatory disorders, post-thrombotic and congenital varices, absence of an eco-Doppler, performance of other surgical procedures (e.g., radiofrequency), and lack of consent from the patient for taking part in the study.

### 2.2. Groups

Primary VV (non-recurrent varicose veins): patients with trunk varices and saphenofemoral insufficiency. Undergoing surgery for the first time through crossectomy and stripping of the GSV.RVV: patients with recurrence at saphenofemoral level. In all cases, the GSV had been previously removed by crossectomy + stripping. The patients again undergo surgery with preoperative echography data suggestive of neo-junction and insufficiency of this level (previous saphenofemoral junction).

### 2.3. Study Variables

(a)Data recording log.(b)Samples of venous walls:(b-1)Proximal vein (saphenofemoral junction). The following genes have been identified: endoglin (Eng), vascular endothelial growth factor (VEGF-A), VEGF receptors (VEGFR1 or FLT1 and VEGFR2 or FLK), HIF-1A (hypoxia inducible factor).(b-2)Distal vein (ankle). A fragment of a GSV (Non-recurrent VV group) and varicose vena other than the GSV (RVV group). The following genes have been identified: Eng, VEGF-A, and HIF-1A.(c)Blood samples. An analysis has been conducted of the following soluble molecules in patient plasma: soluble endoglin (sEng), VEGF-A, and soluble VEGF receptors (sFLT1 and sFLK).

### 2.4. Sample Collection, Processing and Storage

Samples of serum and tissues were collected with prior consent. The sample of blood serum was obtained at the time of hospital admission (a few hours before surgery). These samples were processed and stored at −80 °C in the Biobank at the Biomedical Research Institute of Salamanca (IBSAL, Salamanca, Spain), pursuant to the provisions of Spain’s Royal Decree 1716/2011 on Biobanks.

During surgery, a piece of the GSV was removed from the saphenofemoral junction and another from the ankle (according to groups). These pieces were immediately placed in liquid nitrogen and stored at −80 °C for their subsequent analysis.

### 2.5. Analysis of Tissue Samples

RT-qPCR was used to analyse gene expressions related to angiogenesis in the tissue samples: this involved grinding the frozen tissue sample and extracting its RNA. The next step involved retrotranscribing the RNA to obtain the cDNA used to study the gene expression through real-time PCR.

RNA was extracted from the tissue obtained during surgery through the use of the NucleoSpin^®^ RNA ((Macherey-Nagel, Düren, Germany) commercial kit. The first step involved grinding the previously frozen tissue, taking 20 mg in weight and lysing it with the RA1 reagent supplemented with 1% β-mercaptoethanol using Kimble™ Kontes™ Pellet Pestle™ microcentrifuge tubes. Once the sample has been lysed, the manufacturer’s instructions are followed. The RNA obtained is quantified using a NanoDrop^®^ ND-1000 spectrophotometer (Marshall Scientific), and stored at −80 °C.Obtaining cDNA: the analysis of the gene expression requires using DNA as a reaction substratum, whereby the RNA needs to be converted into DNA via retro or reverse transcription. Viral reverse transcriptase, discovered in 1970, synthesises cDNA from RNA by diluting 250 ng of RNA in each sample with ultrapure water to a final volume of 16 μL. This is followed by the addition of 4 μL of iScript^TM^ Reverse Transcription Supermix (Bio-Rad, Hercules, CA, USA) 5X and incubation in a MyCycler^TM^ (Bio-Rad, Hercules, CA, USA) thermal cycler at 25 °C for five minutes, followed by 30 min at 42 °C, and five final minutes at 85 °C. The cDNA obtained is stored at −20 °C.Quantitative PCR, qPCR or PCR in real time: the PCR is prepared with a final volume of 20 μL:1 μL of cDNA, 10 μL of Supermix iQTM SYBR^®^ Green (Bio-Rad, Hercules, CA, USA), 0.4 μL of each primer at a concentration of 20 mM, and 8.2 μL of ultrapure water. The reaction is undertaken in an iQTM 5 thermal cycler, where it follows a protocol for its incubation at 95 °C for five minutes, followed by 40 thirty-second cycles at 95 °C, 30 s at the optimum annealing temperature, and 30 s at 72 °C. The camera attached to the thermal cycler takes an image of the sample at the end of each cycle, detecting the fluorescent signal, which will be increasingly stronger in step with the higher amount of PCR product.

The expression of the mRNA of the GAPDH gene was used for control or housekeeping purposes.

### 2.6. Analysis of Blood Samples

The samples of serum stored in the Biobank have been used to quantify the concentration of the aforementioned molecules. The studies were conducted using ELISA commercial kits, as per the manufacturer’s instructions. We specifically used a variant of the ELISA test called Multiplexed Fluorometric ImmunoAssay (MFIA) or LUMINEX^®^ technology Luminex Technology Multiplex Assays (Thermo Fisher Scientific). Use has been made of LUMINEX^®^ 200 (R&D Systems) kits.

### 2.7. Statistical Analysis

The values obtained in tissues (gene expression) are represented in box plots that show the median and the 25th–75th percentiles, with whiskers showing the 10th–90th percentiles, and the serum levels (soluble molecules) as the mean ± SEM in pg/mL. The data are presented in the tables as means, and SD and SEM, and *p*-values are indicated. T-tests were used for the analysis between groups, and the D’Agostino-Pearson normality test was applied to the datasets prior to statistical comparations. The statistical result is significant from 0.05. All analyses and graphs were performed using GraphPad Prism version 7.0.0 for Windows (GraphPad Software, San Diego, CA, USA, www.graphpad.com, (accessed on 30 January 2022)). software 

### 2.8. Ethical Responsibilities

The research was approved by the ethical research committee at the Salamanca Health Authority (Comité de Ética de Investigación-CEI-del Área de Salud de Salamanca), complying with ethical standards and best clinical practices (World Medical Association Declaration of Helsinki). CEIm Code: PI 2019 03 204. Date of approval: 1 March 2019.

All the patients gave their written consent to take part in the study. All the data have been kept confidential and encrypted pursuant to the provisions of legislation on personal data protection (15/1999) and biomedical research (14/2007).

## 3. Results

This study involved 43 patients divided into two groups: (a) primary and non-recurrent VV (*n* = 24), and (b) RVV (*n* = 19). Table 1 features the distribution by sexes and ages.

### 3.1. Expression of Markers on the Venous Wall

#### 3.1.1. Proximal Venous Samples (Saphenofemoral Junction)

The RVV group recorded a significantly higher expression of Eng (*p* = 0.0104), VEGF-A (*p* = 0.0074), VEGFR1 (*p* < 0.0001) and VEGFR2 (*p* < 0.0001) compared to the group of primary VV (with no recurrences) (Figure 2). HIF-1A did not record any differences between the groups. Table 2 provides more information.

#### 3.1.2. Distal Venous Samples

None of the three molecules investigated (Eng, VEGF-A and HIF-1A) recorded differences between groups (Figure 3).

### 3.2. Serum Determinations of Soluble Molecules

Of the four molecules studied, only sFLT1 recorded significant differences between the groups (*p* = 0.0392) in terms of a lower concentration in the RVV group compared to the non-recurrent one (Figure 4). More information in Table 3.

There is no correlation between levels of tissue (Eng) and serum (sEng). See Appendix A.

## 4. Discussion

The principles for the surgical treatment of VV were defined in 1950: “to prevent reflux from the deep venous system to the superficial venous system”. Since then, considerable progress has been made in their diagnosis (e.g., preoperative eco-Doppler) and therapy (e.g., endovascular techniques), but RVV have yet to be resolved. There is still some debate over which technique (e.g., stripping, radiofrequency and endovenous laser treatment) is more effective. The RECLAS study attributes the same rate of recurrence to laser treatment as to saphenofemoral ligation/stripping [13]. A subsequent study did not find any differences either in terms of recurrences between standard surgery and endovenous treatment (laser or radiofrequency), although the causes of the relapses are different [14]. Finally, a recent systematic review, endorsed by three major vascular societies, confirms that the high ligation of the saphenous and stripping record similar rates of long-term closure as the latest surgical techniques [15].

Of the four causes of RVV described, neovascularisation is the mechanism most recently involved in the pathogeny of RVV. This is revealed by eco-Doppler [8], quantified by 3D reconstruction images [16], and confirmed by the histopathological analysis of surgical parts [17]. Neovascularisation is a common cause of post-surgical RVV, with the use of eco-Doppler revealing it accounts for 25–94% of these cases [7,8]. Our results indicate that neovascularisation is part of the complex etiopathogenesis of RVV by reporting a significant increase in angiogenesis markers among patients with RVV.

Neovascularisation involves the formation of new blood vessels, which in the case of RVV occupy an abnormal position. These new vessels, of a different sizes, numbers and pathways, appear after both standard surgery (stripping) and endovenous treatment (laser or radiofrequency), albeit less so in techniques of endovenous ablation, given that these procedures do not involve the shedding of endothelial cells, which may be the origin of neovascularisation. In fact, rechannelling is the most common cause of RVV in these preceding techniques [14].

The pathogenesis of neovascularisation, as the cause of post-surgical RVV, considers both intraoperative factors (surgical technique, resulting trauma, suture material, etc.), and postoperative ones (hypoxia, inflammation, pro-angiogenic molecules, etc.).

There are several hypotheses, with the most accepted one being that hypoxia of the venous wall, neovascularisation, and RVV are closely related. Neovascularisation is caused by the angiogenic stimulus in the cicatricial area of the ligation of the saphenofemoral junction and extraction of the saphenous trunk, forming neo-vessels that reconnect with recurrent residual venous trunks. In other words, it may be a response to the venous disconnection; a cicatricial response that through hypoxia triggers endothelial activation and the release of angiogenic factors that produce a rechannelling and endothelisation of the trajectory of the resected vein. The maintenance of the angiogenic signal, characterised by the persistence of angiogenic markers in the neo-vessel, would explain the winding network of neo-vessels that connect the stump of the saphenous vein, its tributaries and the common femoral vein.

Although there are observational studies that correlate the findings of the preoperative eco-Doppler with the macroscopic imprint during the reintervention and histological study of the piece as suggestive of neovascularisation [8], research should focus on the physiopathological mechanisms that cause it following a proper surgical procedure. This means attention should turn to the target molecules involved in neoangiogenic processes, such as HIF, VEGF and endoglin.

HIF has been studied in the pathogenesis of various arterial vascular diseases (e.g., arteriosclerosis) and venous ones of both an acute and chronic nature (VV) [18]. It is well known that the activation of HIF is involved in angiogenesis. In our research, although HIF-1 is higher than in a control group (patients without VV), no significant differences were found between the groups of patients with non-recurrent VV and those with RVV, HIF-2 was not determined. The absence of differences could indicate that the vessel has already formed, and the blood flow has been restored in the area.

As regards the markers of neovascularisation, there is a higher expression of decorin in redundant stumps (after the ligation of the saphenofemoral junction) [10]. For these authors, decorin, a protein of connective tissue, allows differentiating between a redundant stump and a redundant stump + neovascularisation as a cause of RVV. Other markers used (e.g., VEGF, TGF-β1 and 3, metalloproteinase 1) did not record any differences across groups. By contrast, our study reveals significant increases in VEGF-A, its receptors (FLT1 and FLK), and endoglin; with the last of these being a coreceptor of TGF-β.

Endoglin is a membrane glycoprotein that is expressed especially in endothelial cells. It participates in the signalling of different molecules of the TGF-β family. It modulates the cellular responses to TGF-β, including the production of the extracellular matrix, the regulation of angiogenesis, vascular remodelling and cardiovascular development [19]. The presence of endoglin in the cells of the three layers of blood vessels (tunica intima, tunica media, and tunica externa), suggests a major role in vascular physiology. In turn, TGF-β acts in the neoformation of capillaries and in maintaining the integrity of vessel walls, both in the embryo and in postnatal life [12,19]. Besides the membrane form, there is a soluble kind of endoglin (sEng) formed by the proteolytic processing of the membrane isoform [19].

This means that as a pro-angiogenic molecule, endoglin plays a key role in the regulation of neo-angiogenesis, being essential for proper angiogenesis [12]. It has recently been posited that just as an increase in endoglin expression is required for the proper development of the angiogenic process, a decrease in its expression is needed for angiogenesis to be suitably resolved. The persistence of the endoglin expression gives rise to vascular alteration due to excess angiogenesis.

With a view to confirming that the RVV group had greater angiogenesis in the area subject to surgery, we also analysed other markers such as VEGF-A, which is a known and significant factor involved in angiogenesis, and its receptors of types 1 (FLT1) and 2 (FLK). The results confirm that the proximal veins (saphenofemoral junction) in patients with RVV record a significant increase in the expression of all these genes related to angiogenesis. At the same time, it is important to stress that the higher expression, including Eng, is not recorded in the distal areas from the saphenofemoral junction.

The role of endoglin (and its soluble form) has not been studied in the case of RVV. There is only one article that analyses the role of TGF-β and Eng in the cicatrisation of venous ulcers [20]. Our research has studied both forms of endoglin, although unfortunately it has not been able to establish a correlation between them. Along these lines, our study also sought a potential clinical impact. Considering the positive results for endoglin in tissues, we expected similar outcomes for soluble endoglin (sEng), whose determination in plasma (not in tissues) could become a predictive marker of RVV.

Among the soluble molecules studied in the blood, the VEGF type 1 (sFLT1) receptor could be of use, as it is significantly lower in the group of RVV patients; it should be remembered that sFLT1 has an antiangiogenic effect, so a lower quantity makes it easier for neovascularisation to develop.

With a view to reducing or preventing the stimulation of post-surgical angiogenesis, barrier techniques have been proposed—closure of the cribriform fascia, covering of the stump with a PTFE prosthesis-, the reversal of the endothelium of the stump, or avulsion techniques [21,22,23]. These prophylactic techniques are not particularly recommended [24].

Clarifying the role of the different molecules involved in angiogenesis provides insight into one of the most frequent causes of VVR. Together, discovering the “key” molecule in this process could have predictive implications.

Our study has several limitations. First, the low number of patients in each group is due to the reduction in surgical procedures during the COVID-19 pandemic. Second it has not been possible to form a control group (patients without VV) with comparable age and sex characteristics to the VV group. Third, the limited number of molecules analysed is due to cost issues; we have chosen those that have not previously been studied (Eng), and others that have indeed been investigated are reference templates (VEGF and HIF). Fourth, it is not possible in methodological terms to conduct a randomised study, although the biological analyses of the sample have been blinded.

## 5. Conclusions

To conclude, we have found an increase in angiogenesis markers in patients with VV. Specifically, the venous wall at saphenofemoral level in these patients (compared to the group with no recurrences) records a higher and significant expression of Eng, VEGF-A and its receptors FLT1 and FLK. We did not find any differences regarding HIF-1A. Further studies are called for to discover whether the decrease in sFLT1 recorded in patients with a recurrence might serve a predictive purpose.

## Figures and Tables

**Figure 1 jpm-12-00528-f001:**
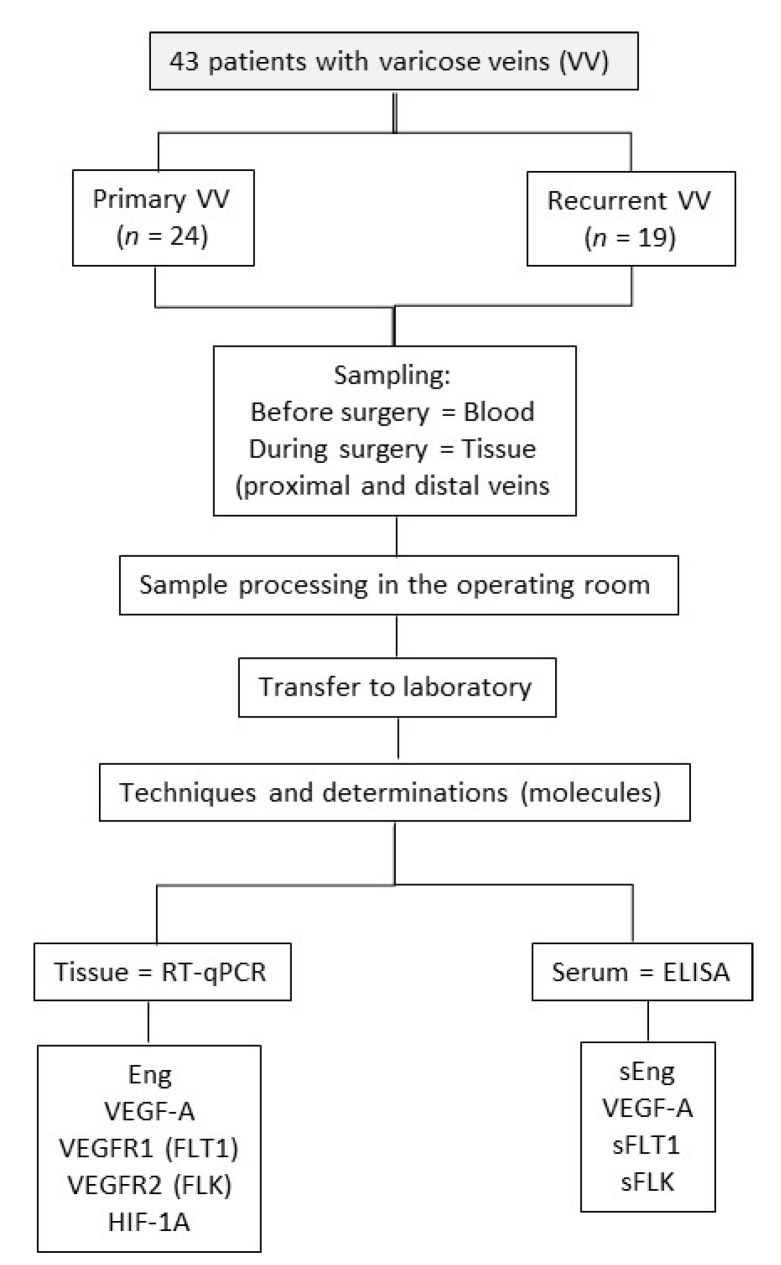
Scheme of the investigation. RT-qPCR, reverse transcription polymerase chain reaction; ELISA, enzyme-linked immunosorbent assay; Eng, endoglin; VEGFA, vascular endothelial growth factor A; VEGFR1, vascular endothelial growth factor receptor 1 (or FLT1); VEGFR2, vascular endothelial growth factor receptor 2 (or FLK), HIF1A, hypoxia-inducible factor 1A. Note: FLT1 and FLK were not determined in the samples of distal veins.

**Figure 2 jpm-12-00528-f002:**
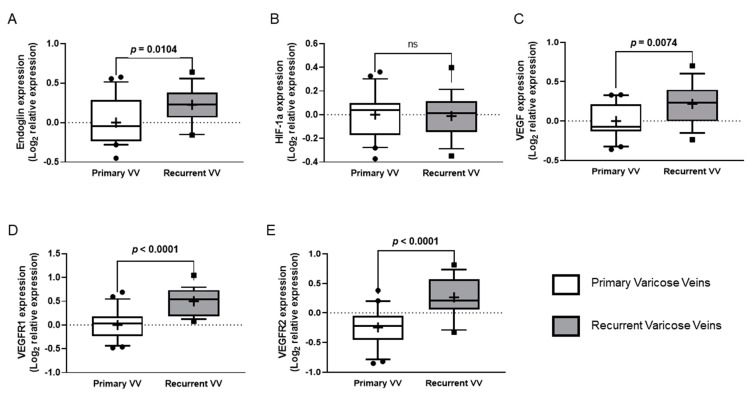
Expression of angiogenesis-related genes in proximal samples. (**A**–**E**) Quantitative analysis of expression of the genes investigated in samples of the proximal great saphenous vein (tissue obtained during surgery) was performed by quantitative RT-PCR. Relative expression of Eng: endoglin (**A**), HIF1A: hypoxia-inducible factor 1A (**B**), VEGFA: vascular endothelial growth factor A (**C**), VEGFR1: vascular endothelial growth factor receptor 1 (**D**) and VEGFR2: vascular endothelial growth factor receptor 2 (**E**) in recurrent varicose vein vs primary varicose vein were analysed. For data normalisation, qPCR analyses are represented as logarithmic transformation of the relative expression; box plots show median and 25–75 percentiles, and whiskers show 10–90 percentile. The *p*-value obtained after t-test analysis indicated if it is statistically significant. ns: not significant.

**Figure 3 jpm-12-00528-f003:**
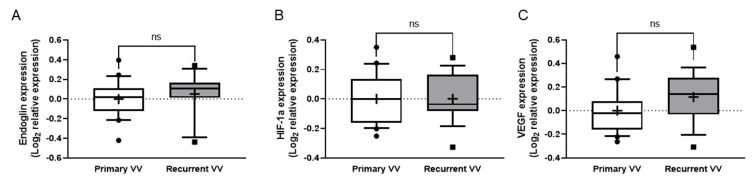
Expression of angiogenesis-related genes in distal samples. (**A**–**C**) Quantitative analysis of expression of the genes investigated in samples of the distal great saphenous vein or other vein -according to groups- (tissue obtained during surgery) was performed by quantitative RT-PCR. Relative expression of Eng, endoglin (**A**), HIF1A: hypoxia-inducible factor 1A (**B**) and VEGFA: vascular endothelial growth factor A (**C**) in recurrent varicose vein vs primary varicose vein were analysed. For data normalisation, qPCR analyses are represented as logarithmic transformation of the relative expression; box plots show median and 25–75 percentiles, and whiskers show 10–90 percentile. ns: Not significant.

**Figure 4 jpm-12-00528-f004:**
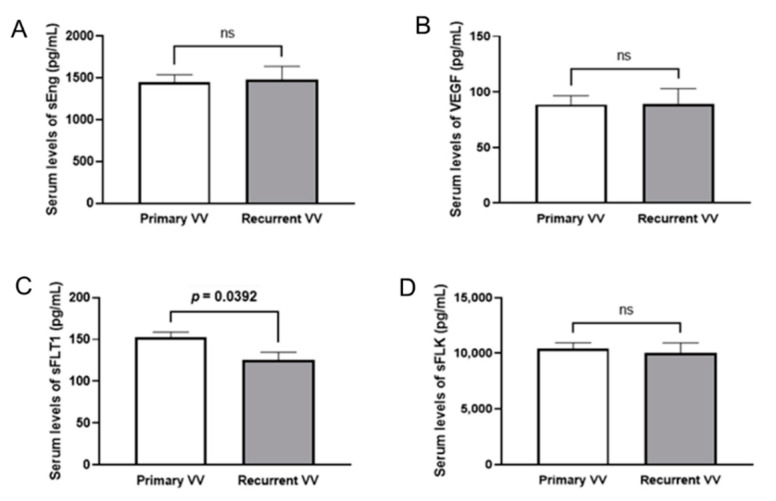
Serum levels of angiogenic or anti-angiogenic soluble factors. (**A**–**D**) Quantitative analysis of the concentration (pg/mL) of the molecules investigated in the serum of patients (obtained before surgery), sEng: soluble endoglin, (**A**), VEGFA: vascular endothelial growth factor A (**B**), sFLT1: soluble fms-like tyrosine kinase 1 (or VEGFR1) (**C**) and sFLK: soluble foetal liver kinase (or VEGFR2) (**D**) were analysed in both patients with primary varicose vein and patients with recurrent varicose vein. Mean ± SEM are displayed. The *p*-value obtained after t-test analysis indicated if it is statistically significant. ns: not significant.

**Table 1 jpm-12-00528-t001:** Basic characteristics of groups.

Group	Patients	Sex (M/F) *	Age (Years)
Primary Varicose Vein	24	7/17	33–67
Recurrent Varicose Vein	19	8/11	32–69

* M (male); F (female).

**Table 2 jpm-12-00528-t002:** Expression (%) of the genes investigated in samples of the proximal internal saphenous vein.

Primary Varicose Veins				Recurrent Varicose Veins			
Genes *	Mean	SD	SEM	Mean	SD	SEM	*p*-Value
Eng	100	77.65	16.94	149.8	75.29	17.75	0.0104
HIF1A	100	44.14	96.32	97.28	43.94	10.38	0.8554
VEGFA	100	50.13	10.94	173.0	104.1	24.54	0.0074
VEGFR1	100	88.50	19.31	287.7	186.3	43.91	< 0.0001
VEGFR2	100	80.37	17.54	346.4	255.1	60.13	< 0.0001

* Genes investigated: Eng, endoglin; HIF1A, hypoxia-inducible factor 1A; VEGFA, vascular endothelial growth factor A; VEGFR1, vascular endothelial growth factor receptor 1 (FLT1); VEGFR2, vascular endothelial growth factor receptor 2 (FLK).

**Table 3 jpm-12-00528-t003:** Serum levels (pg/mL) of the molecules investigated.

Primary Varicose Veins				Recurrent Varicose Veins			
Molecules *	Mean	SD	SEM	Mean	SD	SEM	*p*-Value
sEng	1450	624.0	87.38	1476	629.7	162.6	0.8857
VEGFA	88.66	54.78	7.826	89.41	52.57	13.57	0.9630
sFLT1	152.6	45.53	6.314	125.8	34.75	8.973	0.0392
sFLK	10419	3806	527.7	10018	3544	915.0	0.7166

* Molecules investigated: sEng, soluble endoglin; VEGFA, vascular endothelial growth factor A; sFLT1, soluble fms-like tyrosine kinase 1 (sVEGFR1); sFLK, soluble foetal liver kinase (sVEGFR2).

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
