# Peer review of "Endoglin and Other Angiogenesis Markers in Recurrent Varicose Veins"

_jpm, 2022, doi:10.3390/jpm12040528_

Round 1

Reviewer 1 Report

  1. English language editing is required for phrases construction, grammatical and some spelling mistakes.
  2. All abbreviations must be mentioned in full in their first mention in text, then the abbreviated words are mentioned between brackets. This should be performed throughout the manuscript.
  1. Conclusion in the abstract: It should be clarified that the results for endoglin and other markers are increased in proximal vein samples.
  2. References: Some references are not consistent, names of the same authors are mentioned in different ways.
  3. Great Saphenous Vein (GSV) instead of Internal Saphenous Vein: The recent nomenclature of anatomical parts should be followed and should be unified throughout the manuscript.
  4. In recurrent cases, the GSV may not be previously removed at the ankle level. How could you prove that all lower vein samples were from recurrent GSV in all cases? If it was not previously removed at this level, the veins are thus not recurrent. This gives the lower vein samples little significance and great bias.
  5. Which types of operations were previously performed for the recurrent cases?
  6. The authors consider 4 types of recurrence in the Intro, however, in methods they ignored describing the causes of recurrence in their cases. As the theory in this manuscript resides in investigating markers of neovascularization as a cause of recurrence, therefore, it is important to differentiate recurrence due to possible neovascularization, which should be included, from recurrences from technical or tactical errors, most of which should not be included in the study of neovascularization.
  7. In tables 2 and 3 and figure 2: Data are non-normally distributed, however, the analysis was performed using mean and SD. These data sets should be expressed as median and range. Was a test for normality performed? Why all the expressions of genes in primary group are 100%? P-values are better to be added to tables.
  8. Section 3.1.2. It is mentioned that there are only three investigated molecules, while in patients and methods, the number of studied variables in each of proximal and distal venous segments is 5. In serum, the investigated molecules are 4 not 5 as mentioned in section 3.2.
  9. Page 8 of 11, higher 4 lines: the authors mention that new vessel formation is equal after all venous surgeries, then in the same paragraph they mention the opposite. Obviously, the first statement is not accurate.
  10. Put another way: What does this mean?
  11. "The absence of differences could be indicative that the vessel has already formed and blood flow has been recovered in the area." Please explain clearly this statement.
  12. Control group with no VVs can be obtained from patients undergoing coronary bypass or peripheral arterial bypasses with saphenous grafts.
  13. In figure 5, the hypothesis: A figure alone is not sufficient for explaining what the hypothesis is. As well, all pathways in this figure which are not proved in the current study require references in the explanation text.

Author Response

Reviewer 1

We should like to thank you for your comments and suggestions.

COMMENT 1: English language editing is required for phrases construction, grammatical and some spelling mistakes.

RESPONSE:

A new version of the entire manuscript has been drafted in English, correcting all grammatical errors and spelling mistakes.

COMMENT 2: All abbreviations must be mentioned in full in their first mention in text, then the abbreviated words are mentioned between brackets. This should be performed throughout the manuscript.

RESPONSE:

The abbreviations have been checked. They are written out in full when they first appear in the text, and from then on they are abbreviated in brackets.

COMMENT 3: Conclusion in the abstract: It should be clarified that the results for endoglin and other markers are increased in proximal vein samples.

RESPONSE:

It is now stated that the results for endoglin and other markers are increased in the proximal vein samples.

COMMENT 4: References: Some references are not consistent, names of the same authors are mentioned in different ways.

RESPONSE:

References: the names of authors have been checked and corrected.

COMMENT 5: Great Saphenous Vein (GSV) instead of Internal Saphenous Vein: The recent nomenclature of anatomical parts should be followed and should be unified throughout the manuscript.

RESPONSE:

The following changes have been made throughout the manuscript (text and tables): Great Saphenous Vein (GSV) instead of Internal Saphenous Vein.

COMMENT 6: In recurrent cases, the GSV may not be previously removed at the ankle level. How could you prove that all lower vein samples were from recurrent GSV in all cases? If it was not previously removed at this level, the veins are thus not recurrent. This gives the lower vein samples little significance and great bias.

RESPONSE:

Indeed, in all the recurrent cases (RVV) the GSV was previously removed by stripping. This important aspect is clarified in the section on materials and methods. The distal sample was obtained from another vein in the ankle area. The aim of studying distal samples was to find out whether they were involved in angiogenesis.

COMMENT 7: Which types of operations were previously performed for the recurrent cases?

RESPONSE:

Crossectomy + GSV stripping were performed in all cases. As noted in the previous point, this aspect is clarified in materials and methods: groups.

COMMENT 8: The authors consider 4 types of recurrence in the Intro, however, in methods they ignored describing the causes of recurrence in their cases. As the theory in this manuscript resides in investigating markers of neovascularization as a cause of recurrence, therefore, it is important to differentiate recurrence due to possible neovascularization, which should be included, from recurrences from technical or tactical errors, most of which should not be included in the study of neovascularization.

RESPONSE:

As specified in the groups section in materials and methods, all the RVV patients recorded an insufficiency in the saphenofemoral junction and images suggesting (eco-Doppler) a neo-junction. All the patients in this group had previously undergone surgery by our team (vascular surgeons with experience in eco-doppler exploration, and surgical techniques and tactics). This is discussed in material and methods: inclusion criteria and groups.

COMMENT 9: In tables 2 and 3 and figure 2: Data are non-normally distributed, however, the analysis was performed using mean and SD. These data sets should be expressed as median and range. Was a test for normality performed? Why all the expressions of genes in primary group are 100%? P-values are better to be added to tables.

RESPONSE:

The data displayed in Figures 2 and 3 show the relative expression of each gene analyzed by SYBR green-qPCR, so the result is the percentage with respect to the control sample. Therefore, as the reviewer rightly points out, the data would, by definition, be non-normally distributed.

Although we kept the graphical representation as % vs normal, the data were normalized for the statistical analysis by calculating the logarithm of the original data. The data therefore already followed a normal distribution, which was verified by the D'Agostino & Pearson test.

In the original version of the paper, we kept the graphical representation in the form of % vs normal because we understood it was more intuitive. Following the reviewer's suggestion, we have changed Figure 2 (and 3) to express the median and range, although now of the normalized data. This means the controls will be around 0 and the values of the RVV group will be positive in the case of overexpression, and negative for underexpression.

In addition, as suggested by the reviewer, we have added the p-value to Tables 2 and 3.

The corresponding section of materials and methods has been modified accordingly.

COMMENT 10: Section 3.1.2. It is mentioned that there are only three investigated molecules, while in patients and methods, the number of studied variables in each of proximal and distal venous segments is 5. In serum, the investigated molecules are 4 not 5 as mentioned in section 3.2.

RESPONSE:

Section 3.1.2. These errors in the materials and methods section (study variables) have been corrected and clarified, as well as in Figure 1. The maximum number of molecules identified were five (on the proximal venous wall); three on the distal venous wall (the two VEGF receptors were not included, as we understood that they would not provide any further information, but would incur a higher cost), and four in serum (HIF-1A is not determined).

COMMENT 11: Page 8 of 11, higher 4 lines: the authors mention that new vessel formation is equal after all venous surgeries, then in the same paragraph they mention the opposite. Obviously, the first statement is not accurate.

RESPONSE:

Page 8 of 11, top four lines: redrafted.

COMMENT 12: Put another way: What does this mean?

RESPONSE:

The phrase “Put another way” has been removed and replaced by a different wording.

COMMENT 13: "The absence of differences could be indicative that the vessel has already formed and blood flow has been recovered in the area." Please explain clearly this statement.

RESPONSE:

The phrase has been redrafted.

COMMENT 14: Control group with no VVs can be obtained from patients undergoing coronary bypass or peripheral arterial bypasses with saphenous grafts

RESPONSE:

We do indeed have a control group of patients that have undergone peripheral arterial surgery (inguinal), but these patients are too old for their comparison with VV patients.

COMMENT 15: In figure 5, the hypothesis: A figure alone is not sufficient for explaining what the hypothesis is. As well, all pathways in this figure which are not proved in the current study require references in the explanation text.

RESPONSE:

We agree with your view. Figure 5 has been removed to avoid confusion.

Reviewer 2 Report

This is a prospective observational study that aims to study the expression of some molecules involved in neovascularization process in patients with recurrent varicose veins.

The subject of the article is relevant and fits JPM's scope.

Some comments:

  1. Title and abstract are ok.
  2. Introduction. Line 36. Which medical-legal problems could be caused by varicose disease?
  3. Line 43. What are the references to the statements made in the last sentence of this paragraph?
  4. Introduction. Paragraphs 3 and 4 contain similar information and seems redundant.
  5. The introduction should contain more information about endoglin, in which tissues it is measured, in which situations its expression is increased, what is the justification for investigating it in this population. These data can influence, for example, the determination of inclusion/exclusion criteria.
  6. Recurrent varicose disease occur not only due to neovascularization, but also due to degeneration of pre-existing vessels, which were not diseased at the time of the first operation.
  7. Introduction. Line 59. According to the authors, the aim of the study is to investigate several molecules, but only endoglin was cited. In the methods, we found that other molecules were studied. I think that pointing them out in the introduction can increase interest in the study.
  8. The objectives are not clear enough. If I am not mistaken, the aim is to study a possible role of neovascularization process as a cause of varicose veins recurrence by comparing endoglin and other molecules levels in a group of patients operated on for varicose veins for the first time with another group of patients with recurrent varicose veins.
  9. Methods. How did the authors estimate the sample size?
  10. Some inclusion criteria are just the opposite of other inclusion criteria. For example, if the inclusion criterion already foresees patients with CEAP 2-6, belonging to CEAP 0-1 is not an exclusion criterion, since these patients were not included.
  11. What the authors mean with "Data Logbook" as a study variables?
  12. Results. The values of each molecule expression should be described directly in the text. Figures 2-4 do not point the exact values.

Author Response

Reviewer 2

Thank you for your comments and suggestions. The following points describe the changes made accordingly.

COMMENT 1: Title and abstract are ok.

RESPONSE:

No change made

COMMENT 2: Introduction. Line 36. Which medical-legal problems could be caused by varicose disease?

RESPONSE:

An increasing number of cases are being reported that involve fatalities caused by pulmonary embolism among patients subject to VV surgery in which the need for antithrombotic prophylaxis is questioned.

COMMENT 3: Line 43. What are the references to the statements made in the last sentence of this paragraph?

RESPONSE:

Line 43. This manuscript does not follow this line. You may be referring to the Biomedical Research Institute (IBSAL).

COMMENT 4: Introduction. Paragraphs 3 and 4 contain similar information and seems redundant.

RESPONSE:

Paragraphs 3 and 4 have been merged, as they do indeed contain similar information.

COMMENT 5: The introduction should contain more information about endoglin, in which tissues it is measured, in which situations its expression is increased, what is the justification for investigating it in this population. These data can influence, for example, the determination of inclusion/exclusion criteria

RESPONSE:

Following your indications, the introduction now includes information on endoglin.

COMMENT 6: Recurrent varicose disease occur not only due to neovascularization, but also due to degeneration of pre-existing vessels, which were not diseased at the time of the first operation.

RESPONSE:

We agree.

COMMENT 7: Introduction. Line 59. According to the authors, the aim of the study is to investigate several molecules, but only endoglin was cited. In the methods, we found that other molecules were studied. I think that pointing them out in the introduction can increase interest in the study.

RESPONSE:

Introduction. The study purpose has now been clarified. Given its novelty in studies of this nature -RVV-, endoglin has been highlighted.

COMMENT 8: The objectives are not clear enough. If I am not mistaken, the aim is to study a possible role of neovascularization process as a cause of varicose veins recurrence by comparing endoglin and other molecules levels in a group of patients operated on for varicose veins for the first time with another group of patients with recurrent varicose veins.

RESPONSE:

The study purpose has been clarified. Indeed, the aim is to study the possible role of the neovascularization process as a cause of RVV by comparing endoglin and other molecule levels in a group of patients undergoing surgery for VV for the first time with another group of patients with RVV.

COMMENT 9: Methods. How did the authors estimate the sample size?

RESPONSE:

Our aim was to recruit 30 patients for each group, but this was not possible due to the COVID-19 pandemic.

COMMENT 10: Some inclusion criteria are just the opposite of other inclusion criteria. For example, if the inclusion criterion already foresees patients with CEAP 2-6, belonging to CEAP 0-1 is not an exclusion criterion, since these patients were not included.

RESPONSE:

The inclusion and exclusion criteria have been revised to avoid repetitions.

COMMENT 11: What the authors mean with "Data Logbook" as a study variables?

RESPONSE:

  1. This has been changed to “Data recording log”.

COMMENT 12: Results. The values of each molecule expression should be described directly in the text. Figures 2-4 do not point the exact values.

RESPONSE:

Results. The exact values are reflected in Tables 2 and 3. At the reviewer's suggestion, instead of presenting them as supplementary data, we have incorporated them into the main text. At the suggestion of the other reviewer, these tables now include the exact p-value.

Round 2

Reviewer 1 Report

No comment

Author Response

Dear Reviewer:

We should like to thank you for your comments and suggestions.

We have revised the manuscript and modified the requested recommendations for improvement.

We thank you very much for your attention,

Yours Sincerely,

Begoña García Cenador Ph.D. and Francisco S. Lozano Sánchez M.D., Ph.D

Department of Surgery. University of Salamanca (USAL).
